# Highly Efficient A-to-G Editing in PFFs via Multiple ABEs

**DOI:** 10.3390/genes14040908

**Published:** 2023-04-13

**Authors:** Qiqi Jing, Weiwei Liu, Haoyun Jiang, Yaya Liao, Qiang Yang, Yuyun Xing

**Affiliations:** State Key Laboratory of Pig Genetic Improvement and Production Technology, Jiangxi Agricultural University, Nanchang 330045, China

**Keywords:** ABE, base editing, PFF, start-codon mutation, RNA editing

## Abstract

Cytosine base editors (CBEs) and adenine base editors (ABEs) are recently developed CRISPR-mediated genome-editing tools that do not introduce double-strand breaks. In this study, five ABEs, ABE7.10, ABEmax, NG-ABEmax, ABE8e and NG-ABE8e, were used to generate A-to-G (T-to-C) conversions in five genome loci in porcine fetal fibroblasts (PFFs). Variable yet appreciable editing efficiencies and variable activity windows were observed in these targeting regions via these five editors. The strategy of two sgRNAs in one vector exhibited superior editing efficiency to that of using two separate sgRNA expression vectors. ABE-mediated start-codon mutation in *APOE* silenced its expression of protein and, unexpectedly, eliminated the vast majority of its mRNA. No off-target DNA site was detected for these editors. Substantial off-target RNA events were present in the ABE-edited cells, but no KEGG pathway was found to be significantly enriched. Our study supports that ABEs are powerful tools for A-to-G (T-to-C) point-mutation modification in porcine cells.

## 1. Introduction

Clustered regularly interspaced short palindromic repeats (CRISPRs)/Cas9 technology, which was established over the last decade [1], has become the most widely used genome-editing tool in the life science fields [2,3]. This technology uses the Cas9 enzyme to introduce double-strand breaks (DSBs) at single-guide RNA (sgRNA)-defined target sites, then triggers the DNA repair pathway via nonhomologous end joining (NHEJ), or via homology-directed recombination (HDR) when a donor template is available [4]. However, repair of DSBs induced by CRISPR/Cas9 was frequently found to cause undesirable on-target results, such as large deletions [5], chromosomal translocation [6] and complex rearrangement [5,7].

In recent years, two types of CRISPR/Cas9-based editing platform, cytosine base editors (CBEs) and adenine base editors (ABEs), were constructed via fusing Cas9 nickase (nCas9) with cytidine deaminase and adenosine deaminase, respectively [8,9]. These two base editors can efficiently convert C to T or A to G in a wide range of organisms, including mammalian cells, embryos, various microbes and plants, without creating DSBs or using repair templates [10,11]. The current data strongly suggest that CBEs and ABEs provide more straightforward and efficient platforms in various fields, such as correcting human pathogenic point mutations, improving economic traits of agricultural livestock and plants, generating animal models, gene silencing via introduction of premature termination codons (PTCs) and start-codon mutation [10,11,12].

Pigs are not only an important resource for meat production but also an ideal animal model for human biomedicines [13]. Genetically modified porcine fibroblasts (PFFs) have been commonly used as donor cells for somatic-cell nuclear transfer [14,15], with the aim to produce pigs with improved economic performance or as disease models. In our previous report, we simultaneously introduced PTCs in three tumor suppressor genes in PFFs via a CBE, demonstrating the simplicity and plausible efficiency of the base editing platform [16]. To date, very few studies have been reported on application of ABEs in PFFs [17,18]. In this study, we implemented multiple ABEs, showing excellent editing efficiencies, in HEK293T cells [9,19,20,21] to examine their A-to-G conversion efficiencies, editing windows and RNA editing rates in PFFs.

## 2. Materials and Methods

### 2.1. Construction of Expression Vectors

The ABE7.10 (Addgene plasmid #102919), ABEmax (Addgene plasmid #112095), NG-ABEmax (Addgene plasmid #124163), ABE8e (Addgene plasmid #138489) and NG-ABE8e (Addgene plasmid #138491) editors were a gift from David Liu, and sgRNA expression vector PGL3-U6-sgRNA-PGK-Puro (Addgene plasmid #51133) was a gift from Xingxu Huang. The sgRNAs targeting the *BMPRⅠB*, *CD163*, *APC* and *APOE* genes were designed by online software (http://crispor.tefor.net/, accessed on 2 June 2020.).

We firstly used individual sgRNAs targeting *BMPRⅠB* (1 sgRNA), *CD163* (1 sgRNA), *APC* (2 sgRNAs) and *APOE* (1 sgRNA) (Figure 1) to achieve A-to-G conversion in PFFs. Each sgRNA oligonucleotide was subcloned into the BsaⅠ site of the PGL3-U6-sgRNA-PGK-Puro vector. Moreover, a link-sgRNA strategy (two sgRNAs in one vector) was designed in this study (Appendix A). In brief, the linked BbsⅠ-Hind Ⅲ sites were introduced into the downstream of the sgRNA scaffolding and upstream of the hPGK promoter of the PGL3-U6-sgRNA-PGK-Puro vector, thus generating a new vector referred to as PGL3-V2.0. In another reconstructed vector (referred to as PGL3-V2.1), the linked BbsⅠ-Hind Ⅲ sites were added into the upstream of the U6 promoter, while the BbsⅠ site was added into the upstream of the hPGK promoter of the PGL3-U6-sgRNA-PGK-Puro vector. Then, the *APC* sgRNAs of sites 1 and 2 were subcloned into the BbsⅠ sites of the PGL3-V2.0 and PGL3-V2.1 vectors, respectively. Finally, the PGL3-V2.0 and PGL3-V2.1 vectors were digested by BbsⅠ-Hind Ⅲ enzymes and BbsⅠ enzymes, respectively, and the digested small fragment from PGL3-V2.1 was subcloned into the linearized PGL3-V2.0 to obtain the link-sgRNA expression vector. An EndoFree^®^ Plasmid Maxi Kit (QIAGEN) was used for plasmid purification.

### 2.2. Cell Culture and Transfection

PFFs were isolated from large white fetuses on day 30 of gestation using Dulbecco’s Modified Eagle medium (DMEM, cat. no. 11995-065; Gibco/Thermo Fisher, Waltham, MA, USA) supplemented with 200 U/mL collagenase type Ⅳ (Sigma-Aldrich, St. Louis, MO, USA). Cells were cultured in DMEM supplemented with 10% fetal bovine serum (ExCell Bio, Shanghai, China) and 1% penicillin–streptomycin (Gibco, Grand Island, NY, USA) at 37 °C with 5% CO_2_ in a humidified atmosphere.

Approximately 1 × 10^6^ PFFs were electroporated per reaction with a customized FF-113 + FF-113 dual-electroporation program using a Lonza 4D-Nuclefector system (Basel, Switzerland). For the single-locus editing strategies, 10 µg of sgRNA expression plasmids, 10 µg of ABE plasmids and 1 µg of pMAX-GFP plasmids (provided in the 4D- Nuclefector^TM^ X kit L) were used in each reaction. For dual-site link-sgRNA editing, 10 µg of expression plasmids, harboring both the sgRNAs for sites 1 and 2 (link-sgRNA in Appendix A), 10 µg of ABE plasmids and 1 µg of pMAX-GFP plasmids, were delivered into the PFFs; for dual-site mixed-sgRNA editing (mix-sgRNA in Appendix A), 5 µg each of the sgRNA plasmids for locus 1 and locus 2, 10 µg of ABE plasmids and 1 µg of pMAX-GFP plasmids were used. Each electroporation reaction was performed in triplicate. After transfection, the cells were cultured for 6 h, then replenished with fresh medium and cultured for another 42 h; then, the cells were put under selection in a medium containing 3.0 µg/mL puromycin for single-locus editing in the selection groups (five ABEs) and dual-site editing or directly collected for single-locus editing in the non-selection groups (ABE8e and NG-ABE8e editors). After two days of selection with puromycin, the cells were recovered in fresh medium (without puromycin) for 2–3 days, then used for DNA extraction or further single-colony isolation. Lysis solution (0.45% NP40 and 6 µg/µL proteinase K) was employed for lysing cells, with heating steps of 56 °C for 1 h and 85 °C for 10 min. The lysate was used as a template for PCR amplification using the primers listed in Appendix A. The PCR products were used for TA cloning-based sequencing with a 3130 XL Genetic Analyzer (Applied Biosystem, Carlsbad, CA, USA).

### 2.3. Off-Target Analysis

The potential off-target sites (OTS) for each sgRNA were predicted by the online software (http://crispor.tefor.net/, accessed on 2 June 2020) that was also used for sgRNA design. The top 10 potential OTS for each sgRNA were amplified using the primer list in Appendix A regarding the ABE-edited cell populations. The PCR products were then subjected to TA cloning-based sequencing, and 50 TA clones were picked for each ABE reaction described in Section 2.2.

### 2.4. Isolation of Single-Cell Colonies Targeting the APOE Gene

After the puromycin selection and the recovery steps (described in “Section 2.2”), the transfected PFFs with the plasmid mixture (targeting the *APOE* gene) were seeded into 10 cm petri dishes with various cell densities. After 8–10 days of culturing, single-cell clones on the 10 cm petri dishes were collected and cultivated in 24-well plates. After reaching 80% confluence, about 85% of each cell clone was subcultured, and the remaining cells were used to extract DNA in a lysis solution (0.45% NP40 and 6 µg/µL proteinase K). Then, PCR amplification was performed for the lysate using primers for the *APOE* gene (Appendix A), and the PCR products were sequenced with a 3130XL Genetic Analyzer (Applied Biosystem, Waltham, MA, USA).

### 2.5. Detection of APOE Expression via qRT-PCR and Western Blotting

The total RNA from the single-cell clones carrying mutated start codons (ATG to GTG) and WT cells were extracted using the Trizol Reagent (Invitrogen, Carlsbad, CA, USA) according to the manufacturer’s instructions. The isolated RNA was reverse-transcribed into cDNA using the PrimeScript^TM^ RT Reagent Kit with a genomic DNA Eraser (Takara, Japan), and mRNA expression levels of *APOE* were quantified via the qRT-PCR using the primers listed in Appendix A with a TB Green^®^ Premix Ex Taq^TM^ Ⅱ Kit (Takara, Kyoto, Japan) on an ABI 7900HT Fast Real-Time PCR system (Applied Biosystems, Carlsbad, NM, USA). Porcine *β-actin* was used as an internal control. The relative mRNA expression was determined using the 2^−ΔΔCT^ method. Each reaction was performed in technical triplicate.

The total protein was extracted from the single-cell clones with a RIPA lysis buffer with 1 mM phenylmethylsulfonyl fluoride (PMSF, Beyotime, Shanghai, China) on ice. A bicinchoninic acid (BCA) assay was performed to measure the protein concentration. Ten μg of protein from each sample was resolved using 12% sodium dodecyl sulfate–polyacrylamide gel electrophoresis (SDS-PAGE) (Angle Gene, Nanjing, China), and the separated proteins in the gel were transferred onto a polyvinylidene fluoride (PVDF) membrane (Beyotime, Shanghai, China). Next, the membrane was blocked with Quick Block^™^ Blocking Buffer (Beyotime, Shanghai, China) for 1 h at room temperature, then washed in Tris-Buffered Saline with 0.1% Tween 20 (TBST, Cwbio, Beijing, China) and incubated overnight at 4 °C with a primary antibody against APOE (1:1000; ab52607, Abcam, UK) or β-actin (1:1000; ab8227, Abcam, Cambridge, UK). Finally, the membranes were washed and incubated with a horseradish peroxidase (HRP)-conjugated secondary antibody (1:10,000; ab6721, Abcam, Cambridge, UK) for 1 h at room temperature, and the bands were visualized with BeyoECL Plus (Beyotime, Shanghai, China) in a ChemiDoc^TM^ MP Imaging System (Bio-Rad, Hercules, CA, USA).

### 2.6. RNA Editing Analysis in APOE Gene-Edited PFF Cells

RNA editing analysis was carried out on *APOE* gene-edited cells in this study. After electroporation, the cells were plated on 6 cm petri dishes and cultured for 36 h without puromycin selection. The total RNA from the WT and electroporated cell populations were extracted using the Trizol Reagent (Invitrogen, Carlsbad, CA, USA) according to the manufacturer’s instructions; then, RNA-seq was carried out by Novogene BioTech Co. (Beijing, China) as described in our previous study [16]. Briefly, mRNA was isolated from the total RNA using poly-T oligo beads. The isolated mRNA was fragmented and then reverse-transcribed to yield first-strand cDNA with M-MuLV Reverse Transcriptase using a random hexamer. Second-strand cDNA synthesis was subsequently generated with RNase H and DNA Polymerase I. After adenylation of the ends, ligation of the adapters and PCR amplification, the constructed cDNA libraries were purified with the AMPure XP system (Beckman Coulter, Beverly, USA) with the aim to select cDNA fragments of preferentially 370~420 bp in length. Finally, library quality and integrity were evaluated with the Agilent Bioanalyzer 2100 system, and sequencing was carried out on an Illumina 6000 platform.

Raw data (raw reads) of the fastq format were firstly processed through in-house perl scripts. Then, clean data (clean reads) were obtained via removing reads containing adapters or poly-N and low-quality reads (>50% of the bases had Phred quality scores ≤ 20). At the same time, the Q20, Q30 and GC contents of the clean data were calculated. All the downstream analyses were based on the clean data with high quality. Paired-end clean reads were mapped to the porcine reference genome (Sscrofa11.1; http://asia.ensembl.org/Sus_scrofa/Info/Index, accessed on 10 September 2022.) using Hisat 2 software (version 2.0.5) [22]. The fragment per kilobase of transcript per million mapped fragments (FPKM) of each transcript was calculated using FeatureCounts (version 1.5.0-p3) software [23].

RNA editing sites were identified by REDItools [24], filtered for sites with read coverages ≥ 10, quality scores ≥ 30, homoploymer lengths ≥ 5, editing frequencies ≥ 0.1 and *p*-values < 0.05. KEGG analysis regarding the genes harboring the RNA editing sites was performed using clusterProfiler (version 3.8.1) [25].

### 2.7. KEGG Enrichment Analysis of APOE Gene-Edited Cells

Differentially expressed genes (DEGs) between the *APOE* gene-edited cells (transfected with ABE8e plasmids) from the RNA-editing detection assay and the WT cells were identified using the DESeq2 package (version 1.20.0) [26]. Genes with Benjamini–Hochberg-corrected *p*-values < 0.05 and with log2 (fold change) values > 1 were defined as differentially expressed. KEGG enrichment analysis of DEGs was carried out using clusterProfiler software (version 3.8.1). A corrected *p*-value threshold of 0.05 was used to denote the significance of the enrichment.

### 2.8. Statistical Analysis

Data are shown as means ± standard deviation (SD) unless otherwise stated. Statistical analysis was determined by the independent sample Student’s *t*-test using GraphPad Prism v.8.0.2 software [27]. A *p*-value lower than 0.05 was considered statistically significant.

## 3. Results

### 3.1. Variable A-to-G Editing Efficiencies of Different ABEs in the PFFs

In this study, we firstly detected the A-to-G conversion rates with puromycin selection at target sites in *CD163*, *BMPRⅠB*, *APC* and *APOE* after delivery of each sgRNA expression plasmid and different ABEs. The desired target positions of the *CD163*, *BMPRⅠB*, *APC* (locus 1), *APC* (locus 2) and *APOE* sgRNAs were A5, A7, A4, A7 and A8, respectively (Figure 1). The results showed variable A-to-G conversion rates within the editing windows using different ABEs in the PFFs (Figure 2). For the *CD163* locus, ABE7.10, ABE8e and NG-ABE8e exhibited obviously higher A-to-G editing activity at the A4, A5 and A9 positions than did the ABEmax and NG-ABEmax editors, and ABE8e showed the highest editing activity at this locus among the five editors, achieving averagely >80% of the A-to-G conversion rates at the A4, A5 and A9 positions (Figure 2a). At the *BMPRⅠB* locus, ABE8e and NG-ABE8e achieved significantly higher A-to-G conversion activity than the other three editors at the A2, A7, A9 and A11 positions, while all five editors showed comparable editing efficiencies (>60%) at the A4 position of this locus (Figure 2b); both ABE8e and NG-ABE8e exhibited averagely >60% of the A-to-G conversion rates at all A positions of this locus. At the *APC* loci, the ABEmax and ABE8e editors showed obviously higher editing activity at all adenine positions of locus 1 (Figure 2c), while ABE8e and NG-ABE8e achieved overall superior editing efficiencies to the other three editors at locus 2 (Figure 2d). ABE8e exhibited averagely >80% of the A-to-G editing efficiencies at the A4, A5 and A7 positions of *APC* (locus 1) and around 60% of the A-to-G conversion rates at the A7 position of *APC* locus 2 (Figure 2c,d). At the *APOE* locus, ABE8e and NG-ABE8e showed overall superior editing activity to the other three editors, achieving averagely >80% of the A-to-G conversion rates at all A positions except A11 (Figure 2e). All five ABEs showed broad activity windows: A4-A11 at the *CD163* locus, A2-A11 at the *BMPRⅠB* locus, A1-A12 at *APC* locus 1, A2-A13 at APC locus 2 and A3-A11 at the *APOE* locus. Exceptions were that no editing activity was detected at A11 of the *CD163* locus for NG-ABEmax and that only NG-ABE8e had editing activity at A11 of the *APOE* locus (Figure 2).

The product purity at the target positions of these ABEs varied considerably (Figure 3). The occurrence rates of the desired A-to-G conversions, with or without bystander conversions, were 14.00–92.00% (*CD163* A5), 2.90–88.30% (*BMPRⅠB* A7), 9.33–77.33% (*APC* locus 1, A4), 12.00–61.33% (*APC* locus 2, A7) and 6.30–87.68% (*APOE* A8); however, the precise (desired) A-to-G conversion rates (without bystanders) at these five positions were 2.00–8.67% (*CD163* A5), 0–1.33% (*BMPRⅠB* A7), 0–2.00% (*APC* locus 1, A4), 0.67–12.00% (*APC* locus 2, A7) and 0–5.04% (*APOE* A8) (Figure 3). In other words, high-frequency bystander A-to-G conversions were observed for these target positions (Figure 3, Appendix A).

### 3.2. The Link-sgRNA Strategy Had Superior Editing Efficiencies to the Mix-sgRNA Strategy for Two-site Simultaneous Editing

We compared the efficiencies of two strategies (mix-sgRNA, i.e., two sgRNAs in separate expression vectors; link-sgRNA, i.e., two sgRNAs linked in one vector) in two-site editing of *APC* (Appendix A). The results showed that the link-sgRNA strategy achieved higher editing efficiencies at most A sites (Figure 4a). We then compared the efficiency of simultaneously inducing desired A-to-G conversion (A4 at locus 1 and A7 at locus 2). The data demonstrated that the link-sgRNA strategy achieved significantly higher efficiency (*p* < 0.05) than the mix-sgRNA strategy using the ABEmax editor. When using the ABE8e editor, link-sgRNA also achieved higher efficiency (45.67%) than that of the mix-sgRNA strategy (39.89%), although the difference was not statistically significant (Figure 4b).

### 3.3. Analysis of the DNA Off-Target Effects for Each sgRNA

We examined the top 10 potential OTS in the DNA for each sgRNA in the corresponding ABE-edited cell populations via TA cloning-based sequencing in the PFFs. No DNA off-target editing was found at any of the detected sites (Appendix A).

### 3.4. APOE Expression Was Silenced via ABE-Mediated Start-Codon Mutation

Three single-cell clones with ABE-mediated start-codon mutations (ATG to GTG) in the *APOE* gene were used for qRT-PCR and Western blotting analyses. Two of them carried bystander products (Figure 5a), and no bystander edit beyond the sgRNA region was found. The qRT-PCR results indicated that the mRNA expression of the *APOE* gene decreased by 96% compared to that of the WT cells (Figure 5b). The Western blots demonstrated the absence of APOE protein in gene-edited cells (Figure 5c). Moreover, KEGG enrichment analysis revealed that the DEGs between the *APOE*-edited and WT cells were significantly associated with neurodegenerative disorders such as Alzheimer’s disease, Parkinson’s disease and Huntington’s disease, as well as tumor progression such as melanoma, hepatocellular carcinoma and colorectal cancer (Figure 5d).

### 3.5. All Editors Showed Similar Numbers of RNA Editing Sites and Similar Editing-Type Distribution

We further detected the RNA editing activities of these five ABEs via RNA-seq, from which totals of 62,207, 56,277, 61,274, 53,584 and 39,751 RNA editing sites were identified in the PFFs edited by ABE7.10, ABEmax, NG-ABEmax, ABE8e and NG-ABE8e, respectively. Some randomly selected RNA editing sites are shown in Appendix A. Similar editing-type distributions were found among these five editors, and the RNA editing sites were mostly located in the intron regions (>70%), followed by the 3’ UTRs, the CDS regions, the intergenic regions and the 5’ UTRs (Figure 6a). A total of 12 types of RNA-editing event were identified, of which A-to-G, T-to-C, G-to-A and C-to-T were predominant (Figure 6b). The distribution of the RNA editing types was similar for these five ABEs (Figure 6b). The KEGG-pathway enrichment analysis of the genes involved in RNA editing did not identify a significant enriched pathway (Figure 6c).

## 4. Discussion

CRISPR-based base editors, including CBEs and ABEs, are proven powerful tools that can introduce point mutations at desired sites in various mammalian cell lines [28,29,30], exhibiting variable conversion efficiencies at different targeting sites in mammals [17,31,32]. In the present study, we implemented ABEs including ABE7.10, ABEmax, NG-ABEmax, ABE8e and NG-ABE8e to achieve A-to-G editing at five targeting sites in the *CD163*, *BMPRⅠB*, *APC* and *APOE* genes in PFFs. The desired targeting positions for these five targeting sites were A5, A7, A4, A7 and A8, respectively (Figure 1), leading to start-codon mutation in the *CD163* and *APOE* genes, an A746G mutation in *BMPRⅠB* that significantly affects the ovulation rate in Booroola Merino sheep [33], and amino-acid substitutions in *APC*. Our data revealed that these ABEs had vastly variable editing efficiencies at these five targeting sites (Figure 2). For example, ABE7.10, ABE8e and NG-ABE8e achieved superior A-to-G conversion efficiencies at the A4 and A5 positions of the *CD163* sites when compared with the ABEmax and NG-ABEmax editors, and ABE8e also showed high editing activity at the A9 position of this site (Figure 2a). In addition, though the observed scopes of editing windows varied in different studies [34], the editing windows of these five editors generally fell to A4–A8 in the targeting regions [19,20,34]. In this present study, we obtained 10.00–71.33%, 28.67–76.00%, 9.33–98.00%, 27.33–92.00% and 12.00–88.40% of the A-to-G conversion rates at the A4-A8 positions of these five regions via ABE7.10, ABEmax, NG-ABEmax, ABE8e and NG-ABE8e, respectively (Figure 2). These observed editing efficiencies are comparable to those obtained in mouse zygotes (10.1–88.7%, via ABE7.10) [35], in monkey embryos (20.0–88.6%, via ABEmax) [36] and in HEK293T cells (more than 80%, via ABE8e and NG-ABE8e) [21]. ABE7.10 and ABEmax achieved 32.67% and 47.33% of the A-to-G conversion rates at the *BMPRⅠB* A7 position: significantly higher than those obtained in sheep fibroblasts with ABE7.10 and ABEmax (6.25% and 23.08%, respectively) [37]. However, this significant elevation of editing efficiency might be a result of the 48h puromycin selection after electroporation. Indeed, a comparative experiment of this study revealed that the A-to-G editing efficiencies of ABE8e and NG-ABE8e in the non-selection groups were significantly lower (averagely decreased by 50.04% and 32.82%) than those in the puromycin selection groups (Figure 2 and Appendix A). In general, ABE8e and NG-ABE8e showed overall higher A-to-G conversion rates than the other three editors in the PFFs. Thus, this study suggests that ABE8e and NG-ABE8e are preferred choices for gene silencing because bystander editing is most likely inconsequential with them [34].

We further analyzed the product purity at these five targeting sites, and the overall finding was consistent with previous investigations in mammals [12,36,38]. Variable indel frequencies (0–19.30%) were observed at these five targeting sites; no indel was found at the *CD163* site by any of the editors, and as high as 19.30% was found at *BMPRⅠB* (A7) via the ABE8e editor (Figure 3). No A to non-G conversion was found at any of these five targeting regions. High percentages of bystander A-to-G edits within the activity windows were similar to the findings in our previous CBE experiments in PFFs [16]. It should be noted that although high-frequency bystander A-to-G conversions were found, there was still a small proportion of desired A-to-G edits without bystander products, which mostly were from editors with low A-to-G conversion rates (Figure 3). This suggests that if bystander mutations are not allowed in base editing experiments, then base editors with lower editing efficiencies are preferable. Further engineering efforts have been made to modify the substrate affinities of deaminase domains in order to narrow editing windows and hence to effectively reduce bystander products [39,40,41]. In addition, the results from our link-sgRNA strategy (two sgRNAs expressed by one vector, Figure 4 and Appendix A) support the implication that cloning of multiplex sgRNAs into a single vector can be an effective platform for simultaneously targeting multiple genome regions in mammalian cells [42,43].

ABE-mediated start-codon mutation has been proven to be a practical method for gene silencing [44]. In this study, the start codons of the *CD163* and *APOE* genes were converted into GTG (Figure 1). *APOE* gene knockout was confirmed by Western blotting (Figure 5c), whereas mRNA expression was unexpectedly extremely suppressed (Figure 5b). It is well-known that gene expression is highly associated with histone modifications, such as acetylated H3K27 (H3K27ac) [45]. Our previously reported data [46] indicated that this ABE targeting locus is the H3K27ac enrichment site (Appendix A), and we speculated that start-codon mutation may significantly affect H3K27ac deposition, resulting in dramatically reduced transcription of *APOE* mRNA. The KEGG-pathway analysis regarding RNA-seq data of *APOE* start-codon-mutated cells showed that the enriched pathways were mostly involved in tumor development, cell-cycle progression and neuronal degenerative disease-related pathways (Figure 5d), which is highly consistent with the reported function of the *APOE* gene [47,48,49]. Knockout of *CD163* could not be confirmed in the PFFs, as this gene is primarily expressed in some macrophage cell lineages [50].

Off-target DNA and RNA editing are some of the major concerns about base editing techniques [51]. In this study, no DNA off-target event was found, though only the top 10 potential OTS were identified (Appendix A); however, we detected a large number of RNA off-target events induced by these five editors in *APOE* gene-edited cells, mostly in intron regions, and approximately a quarter of the editing events were A–to–G and C–to–T (Figure 6b). We speculate that relatively high plasmid dosage could be a factor increasing the number of RNA editing sites. It should be noted that the KEGG-pathway analysis in this study exhibited no significant enriched pathway from off-target RNA editing (Figure 6c), suggesting that the side effects of off-target RNA editing could be cautiously optimistic. Wide-spread RNA editing has been a persistent drawback of base editors [39,52], and to minimize the side effect of unwanted RNA editing, engineered deaminases have been tested and significant reduction in off-target RNA editing events could be achieved [39,52]. In addition, research has revealed that delivery of ABE ribonucleoprotein complexes (RNPs) or mRNA can dramatically reduce random RNA editing events, exhibiting more robust strategy for base editing in mammalian cells [53,54].

## 5. Conclusions

In this study, we efficiently generated A-to-G conversions at several loci in PFFs via multiple ABEs, whereas variable editing efficiencies and activity windows were observed. The strategy of two sgRNAs in one vector showed superior editing efficiency to the use of two separate vectors. No DNA off-target event was identified, but a large number of RNA off-target sites were observed.

## Figures and Tables

**Figure 1 genes-14-00908-f001:**
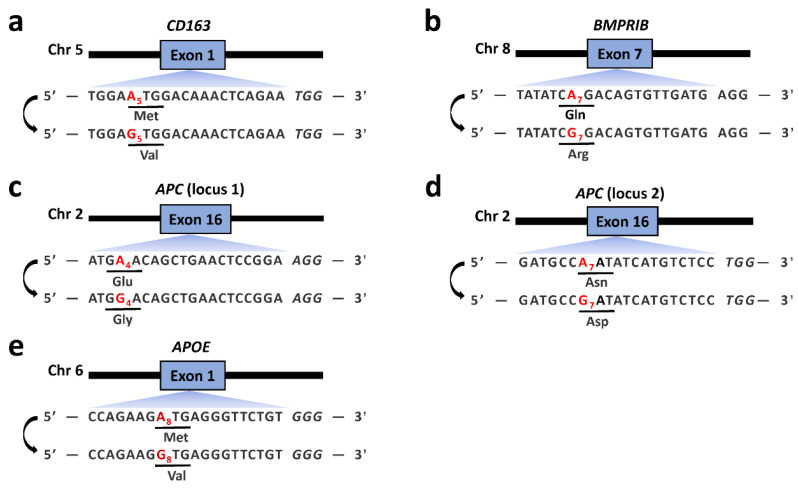
The target sequences at *CD163* (**a**), *BMPRⅠB* (**b**), *APC* (locus 1) (**c**), *APC* (locus 2) (**d**) and *APOE* (**e**). The desired A-to-G conversion at each locus is marked in red.

**Figure 2 genes-14-00908-f002:**
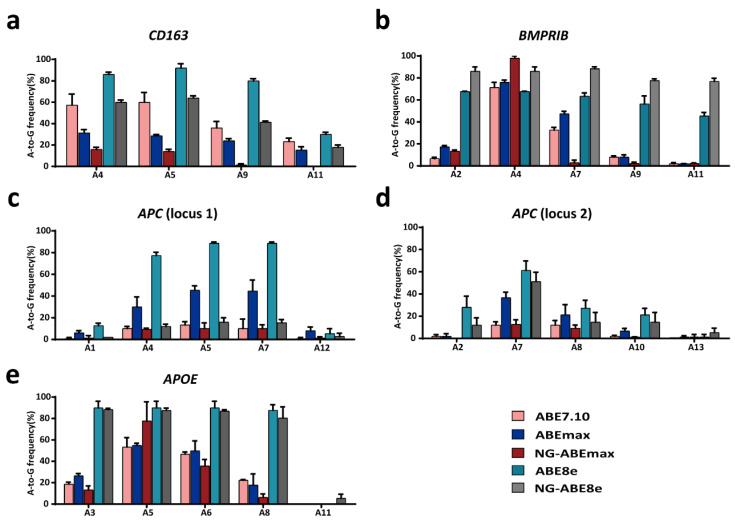
ABE-mediated A-to-G conversion efficiencies at the *CD163* (**a**), *BMPRⅠB* (**b**), *APC* (locus 1) (**c**), *APC* (locus 2) (**d**) and *APOE* (**e**) gene loci with puromycin selection in the PFFs and using multiple ABEs. *-n* = 3, independent replicates.

**Figure 3 genes-14-00908-f003:**
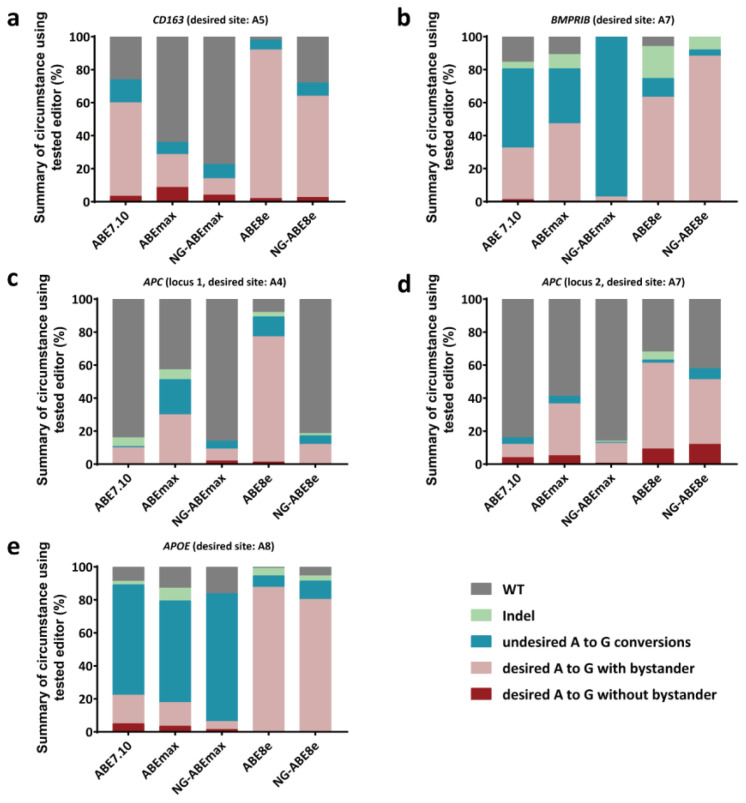
Frequencies of alleles carrying different types of mutations at the *CD163* (**a**), *BMPRⅠB* (**b**), *APC* (locus 1) (**c**), *APC* (locus 2) (**d**) and *APOE* (**e**) gene loci with puromycin selection and using multiple ABEs. *n* = 3, independent replicates.

**Figure 4 genes-14-00908-f004:**
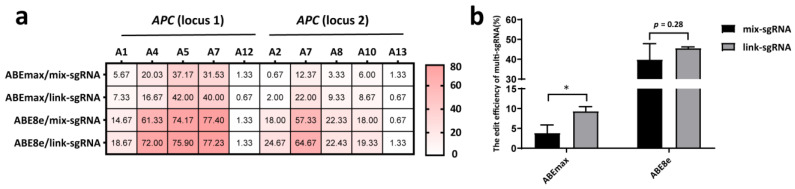
Simultaneous A-to-G conversions at two *APC* loci in the PFFs with the ABEmax and ABE8e editors using the mix-sgRNA and link-sgRNA strategies. (**a**) The heatmap shows the editing efficiencies of each A site at the two targeting loci via the mix-sgRNA and link-sgRNA strategies. (**b**) The efficiencies of simultaneously inducing desired A-to-G conversion at both loci via the mix-sgRNA and link-sgRNA strategies. * *p* < 0.05, *n* = 3, independent replicates.

**Figure 5 genes-14-00908-f005:**
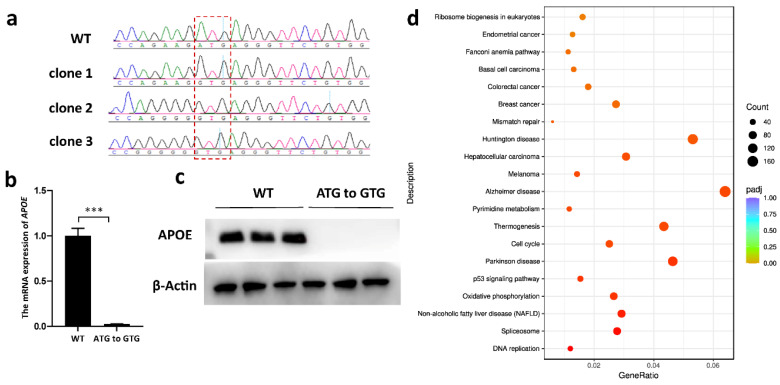
Gene-expression detection and KEGG enrichment analysis in *APOE* start-codon-mutated PFFs. (**a**) Sequencing chromatograms of *APOE*-edited single-cell clones. The start codon is marked by the red frame. (**b**) The relative expression levels detected by qRT-PCR in three *APOE*-edited single-cell clones. *** *p* < 0.001, n = 3, independent replicates. (**c**) A Western blot showing the expressions of APOE in WT and *APOE*-edited PFFs. (**d**) The top 20 enriched KEGG pathways between the *APOE*-edited cell population and the WT PFFs. The *x*-axis represents the enrichment factor. The *y*-axis corresponds to the KEGG pathways. The colors of the dots represent the adjusted *p*-values, and the sizes of the dots represent the number of DEGs mapped to each reference pathway. R package ggplot2 (3.3.1) was performed to draw the bubble chart of the KEGG enrichment analysis.

**Figure 6 genes-14-00908-f006:**
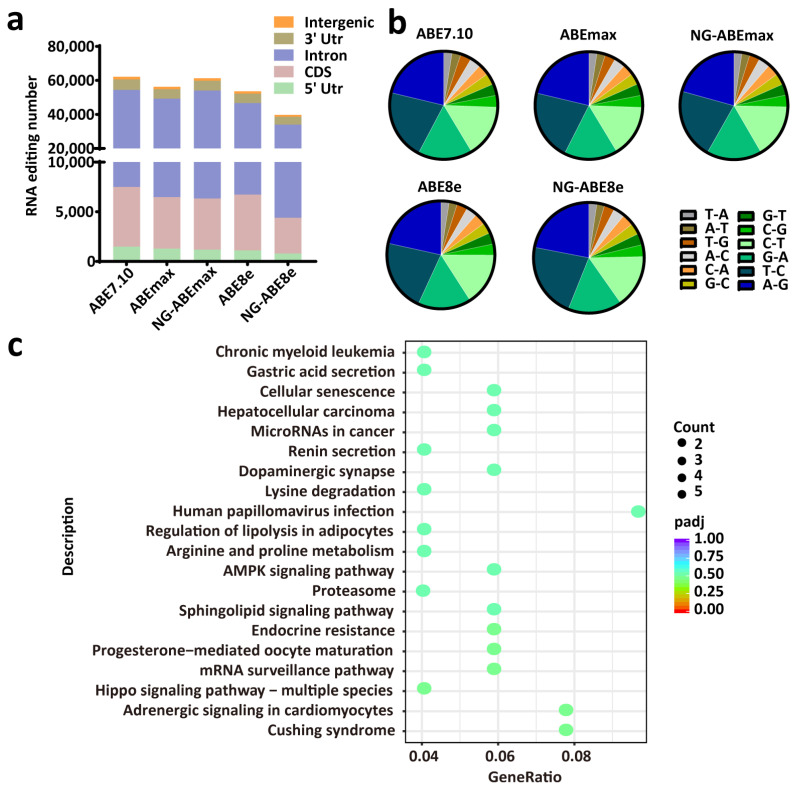
RNA editing activities induced by ABEs in *APOE-*gene-edited PFFs. (**a**) The numbers and distributions of RNA editing sites in genomic regions induced by ABEs in the PFFs. n = 3, independent replicates. (**b**) The pie chart shows the frequencies of different types of RNA editing. (**c**) The top 20 enriched KEGG pathways of RNA editing sites induced by ABE8e. The *x*-axis represents the enrichment factor. The *y*-axis corresponds to the KEGG pathways. The colors of the dots represent the adjusted *p*-values, and the sizes of the dots represent the number of edited genes mapped to each reference pathway. R package ggplot2 (3.3.1) was performed to draw the bubble chart of the KEGG enrichment analysis.

## Data Availability

The RNA-seq raw datasets generated and analyzed during the current study are available in the figshare repository. https://figshare.com/s/503efe78f2d4c0d9bc51, accessed on 24 October 2022.

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
