# Peer review of "Highly Efficient A-to-G Editing in PFFs via Multiple ABEs"

_genes, 2023, doi:10.3390/genes14040908_

Round 1
Reviewer 1 Report
Jing et al. described the use of different adenine base editors (ABEs) in porcine fibroblasts (PFFs) as part of the process to improve current disease models or industry development. They have systematically tested editing in 5 sites from 4 relevant genes in embryo manipulation: CD163, BMPRIB, APC and APOE using 5 different ABEs (ABE7.10, ABEmax, NG-ABEmax, ABE8e and NG-ABE8e). They found high variability across the editors and target sites, but consistently ABE8e and NG-ABE8e delivered the higher editing percentages. Additionally, they reported better efficiencies when they delivered two sgRNAs in-one-vector rather than when two vectors were used. Overall, the authors provide new information about adenine editors efficiency in PFFs that will serve to improve the techniques to genetically modified organisms.
Major comments
1. It is critical/useful/insightful to measure the editing efficiency in the absence of puromycin to obtain a “raw” picture of editing. Importantly, there will be cases where selection is not possible and having previous data of editing efficiencies from various editors would be useful to choose the best strategy undergo the experiment.
2. RNA editing is a concern in the experiments performed. It is probably caused by the high amounts of DNA used in the transfection protocol. Have the authors tried lower amounts? Is it available for the authors RNP of mRNA ABEs to test?
Minor comments
1. All gene symbols must me written in italic.
2. Line 48. The authors probably meant to say ABE instead CBE editor.
3. To address line to line variability it would be relevant to perform ABE editing in different PFF lines – if that is the case already, specify in each experiment the number of biological replicates at least in the Figure Legend. I could only find information regarding the number of technical replicates (3 electroporation reactions per sgRNA) in line 90.
Author Response
Response to Reviewer 1 Comments
Major comments
Point 1: It is critical/useful/insightful to measure the editing efficiency in the absence of puromycin to obtain a “raw” picture of editing. Importantly, there will be cases where selection is not possible and having previous data of editing efficiencies from various editors would be useful to choose the best strategy undergo the experiment.
Response 1: Porcine fibroblasts are hard-to-transfect cells. Without a drug selection process, the screening of transfected cells becomes an extremely tedious and laborious process. In our previous studies, we obtained impressive editing efficiencies in PFFs through puromycin selection (Qiao et al., 2020; Yang et al., 2022). Indeed, it is meaningful to know the “raw” genome editing efficiencies (without puromycin selection); however, PFFs may be not a suitable cell line for that purpose.
The references cited here are listed as follows:
Qiao CM, Liu WW, Jiang HY, et al. Integrated analysis of miRNA and mRNA expression profiles in p53-edited PFF cells. Cell Cycle, 2020, 19 (8): 949-959.
Yang Q, Qiao CM, Liu WW, et al. BMPR-IB gene disruption causes severe limb deformities in pigs. Zoological Research, 2022, 43 (3): 391-403.
Point 2: RNA editing is a concern in the experiments performed. It is probably caused by the high amounts of DNA used in the transfection protocol. Have the authors tried lower amounts? Is it available for the authors RNP of mRNA ABEs to test?
Response 2: We thank the reviewer for this comment. The total ABE plasmid dosage used in different reported studies were ranged from 7.5-30 µg (delivered by liposomes or Lonza Nucleofector). In this study, we used 10 µg ABE plasmid, 10 µg sgRNA expression plasmid and 1 µg GFP plasmid referred to our previous study, using 25 µg CRISPR/Cas9 plasmid (Qiao et al., 2020). Wide-spread RNA editing has been reported in previous reports and has been a persistent problem for base editors (Zhou et al., 2019; Grünewald et al., 2019). In this study, we used ABE plasmid at the medium level (10 µg) of reported doses, and this still likely contributed to the large amount of RNA editing. Using lower editor doses most likely will reduce the number of RNA edits. In addition, as the reviewer pointed out, delivery of ABE RNPs or mRNA can significantly reduce the number of RNA editing sites, and this will be very helpful in our future studies. In the revision, we added description regarding the plasmid dosage and comments on using ABE RNPs or mRNAs, please see page12, lines 375-377, 380 and 382-385.
Grünewald J, Zhou R, Garcia S, et al. Transcriptome-wide off-target RNA editing induced by CRISPR-guided DNA base editors. Nature, 2019, 569: 433-437.
Qiao CM, Liu WW, Jiang HY, et al. Integrated analysis of miRNA and mRNA expression profiles in p53-edited PFF cells. Cell Cycle, 2020, 19 (8): 949-959.
Zhou C, Sun Y, Yan R, et al. Off-target RNA mutation induced by DNA base editing and its elimination by mutagenesis. Nature, 2019, 571(7764): 275-278.
Minor comments
- All gene symbols must be written in italic.
Response: We thank the reviewer for careful reading. We have checked the manuscript carefully and made all gene names in italic type.
- Line 48. The authors probably meant to say ABE instead CBE editor.
Response: In line 48, we mentioned our previous report on the application of CBE in PFFs. For clarity of expression, the description of “In our very recent study” was modified to “In our previous report”. Please see page 2, lines 47-48.
- To address line to line variability it would be relevant to perform ABE editing in different PFF lines – if that is the case already, specify in each experiment the number of biological replicates at least in the Figure Legend. I could only find information regarding the number of technical replicates (3 electroporation reactions per sgRNA) in line 90.
Response: We totally agree that the ABE editing efficiency in different PFF cell lines would be variable. In this study, PFFs were isolated from two fetuses that were collected from the same sow, and passage 1 (P1) or P2 PFFs were used for transfection. Thus, the background of PFFs used in this study could be considered as uniform. In our original manuscript, the sample size was indicated in the legend of Figures 4, 5 and 6. In this revision, we uniformly described as “N=3, independent replicates”. Please see lines 224-225, 229-230, 251-252, 274, 295-296 in the revision.

Reviewer 2 Report
The manuscript “Highly efficient A to G editing in PFFs via multiple ABE editors” describes cell-culture base editing using adenine base editors (ABEs) at 5 genomic loci in porcine fibroblast cells (PFFs). The authors report the efficiency of base conversion of A to G nucleotides across the guideRNA target site of the targeted genes. Different efficiencies are reported for the different designed, publicly available base editors. Editors are delivered as DNA vectors in PFFs, selected on puromycin and target sites sequenced by Sanger sequencing. Authors go on to describe desired and undesired mutations at these gene, examining and reporting by-stander mutations. Not surprisingly, in the absence of by-standers, the based editing frequency is low to absent depending on the gene, the site and the editor. Authors go on to examine multiplex-based editing using mixed- or linked-sgRNA delivery. While the authors report dual-editing at the APC locus (1 and 2) the resultant frequency is not statistically significant depending on the base editor platform. Whether this difference holds for other sites targeted was not tested. The authors document loss of expression (RNA steady-state) and protein for the APO gene edit. Further the authors document a surprisingly high-frequency of RNA editing in the PFF cells across all the editing reagent platforms.
Comments to the authors:
1. The reviewer was not clear whether the analysis of base modification was conducted on pooled cells (post-puromycin selection) or on single-cells.
2. The amount of plasmid delivered is on the high end of delivery. Does this account for the by-stander observations, and for the high RNA editing frequency observed as these reagents would be continually expressed from the DNA vectors and maintained due to selection.
3. With that, was the frequency of transfection of the PFF cells influenced by the different editors. What was the frequency of recovering puro-resistant cells with and w/o editors. Was there some preselection of recovering colonies with the different editing platforms?
4. Fig 5, the editors examine by-stander across the guideRNA site. Is there a precedent for only examining across this window? Did the authors examine beyond this 23 base window for base modification?
5. Section 3.2 does not add a significant contribution to the paper. Either simplify or eliminate.
6. Off-target analysis in this paper is unsatisfactory. Electronic off-target analysis has been demonstrated to be insufficient for identifying bona-fide off-targets identified by other more stringent methods. The authors should revisit this section and devise another strategy to examine whether off-targets exist at the DNA level in PFF cells across these editors.
7. RNA editing section is quite interesting. To point number 6 above, whether the base editors modified RNA, DNA or both would be an open question in light of reexamining whether some of these edits detected in RNA are of DNA origin. While compelling based on the numbers reported, and thus unlikely they all originate from DNA modification, the authors should consider reporting some of the sequences of the RNA modifications in Supplemental data.
8. The discussion lacks significant insights for the results presented in the paper. Other than confirming that different base editors demonstrate different efficiencies at different genes with different by-standers in pig cells, the authors should consider discussing more of the novelty or not, of their findings. How to get around the by-standing editing if important? What about RNA-editing is that a concern?
9. Regarding the APO start codon modification, have the authors considered a simpler model the incorporates loss of protein translation of APO due to the lack of an AUG start codon?
Author Response
Response to Reviewer 2 Comments
Point1: The reviewer was not clear whether the analysis of base modification was conducted on pooled cells (post-puromycin selection) or on single-cells.
Response 1: There were three cell treatment strategies in this study. Cell populations after puromycin selection were used for detection of editing efficiency; the collected single colonies after puromycin selection were used for APOE mRNA and protein expression; the cell populations without puromycin selection were used for RNA editing analysis. The corresponding description were shown in lines 85-96 on page 2, lines 109-112 and lines 143-145 on page 3. In the revision, the description in section “2.6. RNA Editing Analysis in APOE Gene-edited PFF Cells” was modified to make the meaning clearer, please see page 3, lines 144-145.
Point 2: The amount of plasmid delivered is on the high end of delivery. Does this account for the by-stander observations, and for the high RNA editing frequency observed as these reagents would be continually expressed from the DNA vectors and maintained due to selection.
Response 2: Thanks to this comment. Indeed, relatively high plasmid dosage may contribute to the bystander products as well as high RNA editing sites. It is well known that bystander products are unavoidable when using available ABE or CBE editors. In this study, these ABE editors exhibited variable activity windows and editing efficiencies in different loci, suggesting the bystander editing might be evitable. Indeed, decrease the plasmid dosage may reduce the bystander products, whereas likely at the expensive of editing efficiency. A better strategy regarding this issue is to develop ABE editors with narrower editing window, which can effectively reduce bystander editing (Jeong et al., 2021; Chen et al., 2023).
Regarding the RNA editing events, we do realized that the relatively high plasmid dosages might have contributed to the high number of RNA editing sites. The total ABE plasmid dosage used in different reported studies ranged from 7.5-30 µg. The ABE plasmid amount (10 µg) in this study was at the medium level of reported dosages, and the amount of RNA editing events was similar to what has been reported (Zhou et al., 2019 ; Grünewald et al., 2019). We are aware of some optimization strategies such as using lower plasmid dosage or using ABE RNPs or mRNA may significantly reduce the off-target RNA editing. In the revision, we added some description regarding the bystaner editing, plasmid dosage and off-target RNA editing. Please see page 11, lines 351-353 and page 12, lines 375-377, 380 and 382-385.
Chen L, Zhang S, Xue NN, et al. Engineering a precise adenine base editor with minimal bystander editing. Nature Chemical Biology, 2023, 19(1): 101-110.
Grünewald J, Zhou RH, Garcia SP, et al. Transcriptome-wide off-target RNA editing induced by CRISPR-guided DNA base editors. Nature, 2019, 569, 433-437.
Jeong YK, Lee S, Hwang GH. et al. Adenine base editor engineering reduces editing of bystander cytosines. Nature Biotechnology, 2021, 39(11): 1426-1433.
Zhou CY, Sun YD, Yan R, et al. Off-target RNA mutation induced by DNA base editing and its elimination by mutagenesis. Nature, 2019, 571(7764): 275-278.
Point 3: With that, was the frequency of transfection of the PFF cells influenced by the different editors. What was the frequency of recovering puro-resistant cells with and w/o editors. Was there some preselection of recovering colonies with the different editing platforms?
Response 3: We thank the reviewer for this remark. If we understood the first question right, the answer is yes, there were differences in transfection effects among 5 ABE editors in PFFs (see the figure listed as below), and the survive rate of cells transfected with ABE8e was significantly lower than the other 4 groups, whereas the transfection efficiency was quite high in this experiment group. It should be noted that the GFP fluorescence intensity does not necessarily represent the transfection efficiency of the ABE editor, but it still tells that there is something different.
Regarding the second question, we did not examine the rate of surviving cells without editor. The working concentration of puromycin was set to 3.0 µg/mL according to our previous experiment, which found that puromycin with 3.0 µg/mL of concentration can kill all PFFs (P0 to P2) in around 4 days. In this study, we performed puromycin selection for two days with a final concentration of 3.0 µg /mL.
For the third question, as the puromycin selection marker is in the sgRNA expression vector, there was no preselection experiments regarding different ABE editors.
Point 4: Fig 5, the editors examine by-stander across the guide RNA site. Is there a precedent for only examining across this window? Did the authors examine beyond this 23 base window for base modification?
Response 4: We thank the reviewer for this comment. Indeed, the bystander product beyond this 23 base window is worthy of attention. Following this comment, we checked the sequence beyond the 23 base window from the 3 single colonies used to detect expression of APOE gene, and no any edits were found (see the figure listed as below). In the revision, the corresponding information was supplemented, please see page 8, lines 260-263.
Point 5: Section 3.2 does not add a significant contribution to the paper. Either simplify or eliminate.
Response 5: Following this comment, we simplified the section 3.2. Figure 4c and the corresponding description was deleted. Please see page 7, lines 241-244 and page 8, lines 245-254.
Point 6: Off-target analysis in this paper is unsatisfactory. Electronic off-target analysis has been demonstrated to be insufficient for identifying bona-fide off-targets identified by other more stringent methods. The authors should revisit this section and devise another strategy to examine whether off-targets exist at the DNA level in PFF cells across these editors.
Response 6: We totally agree that it is not enough to only detect the top predicted OTS. Up to now, there are various assays for DNA off-target detection for CRISPR-based genome editing (Manghwar et al., 2020). Indeed, genome-wide analysis of OTS can provide much more comprehensive information. Whereas detection of top predicted OTS was also widely used in genome editing studies in plant and mammals (Liu Y. et al, 2020; Liu Y. et al, 2021; Song M. et al, 2021; Zong Y. et al, 2022). The main objective of this study is to detect the editing events regarding activity windows in PFFs via multiple ABE editors, thus we only detected the top 10 predicted OTS. In the revision, we modified the description regarding DNA off-target analysis in the Discussion section, please see page 12, lines 372-373.
The references cited here are listed as below:
Liu Y, Li XY, He S, et al. Efficient generation of mouse models with the prime editing system. Cell Discovery, 2020, 6 (1): 27.
Liu Y, Yang G, Huang SH, et al. Enhancing prime editing by Csy4-mediated processing of pegRNA. Cell Research, 2021, 31 (10): 1134-1136.
Manghwar H, Li B, Ding X, et al. CRISPR/Cas systems in genome editing: methodologies and tools for sgRNA design, off‐target evaluation, and strategies to mitigate off‐target effects. Advanced Science, 2020, 7 (6): 1902312.
Song M, Lim JM, Min S, et al. Generation of a more efficient prime editor 2 by addition of the Rad51 DNA-binding domain. Nature Communications, 2021, 12 (1): 5617.
Zong Y, Liu YJ, Xue CX, et al. An engineered prime editor with enhanced editing efficiency in plants. Nature Biotechnology, 2022, 40 (9): 1394-1402.
Point 7: RNA editing section is quite interesting. To point number 6 above, whether the base editors modified RNA, DNA or both would be an open question in light of reexamining whether some of these edits detected in RNA are of DNA origin. While compelling based on the numbers reported, and thus unlikely they all originate from DNA modification, the authors should consider reporting some of the sequences of the RNA modifications in Supplemental data.
Response 7: Following this comment, sequence alignment was carried out between all predicted DNA OTS (totally 203 loci) and RNA editing sites, and no overlap was identified. This suggests that the RNA editing sites were RNA origin other than DNA origin. In the revision, we supplemented partial sequences of the RNA modifications and controls in Figure S5, and one-sentence description was added in section 3.5 (please see page 9, line 285).
Point 8: The discussion lacks significant insights for the results presented in the paper. Other than confirming that different base editors demonstrate different efficiencies at different genes with different by-standers in pig cells, the authors should consider discussing more of the novelty or not, of their findings. How to get around the by-standing editing if important? What about RNA-editing is that a concern?
Response 8: Following this comment, we reorganized the first paragraph of the Discussion section and added some descriptions regarding the bystander editing and RNA editing issues. Please see page 11, lines 316-323, 329-332, 336-339 and 351-353; page 12, lines 375-377, 380 and 382-385.
Point 9: Regarding the APO start codon modification, have the authors considered a simpler model the incorporates loss of protein translation of APO due to the lack of an AUG start codon?
Response 9: We thank the reviewer for this comment. In general, start codon mutations can stop the translation of mRNA, whereas the mRNA transcription may not affected by the AUG mutation. In this study, the protein expression of APOE was abolished as expected due to the AUG mutation. However, the mRNA expression level of this gene was unexpectedly almost eliminated (decreased by 96%). It is an interesting phenomenon and the detailed mechanisms were unclear. Therefore, we made a simple deduction regarding this phenomenon in Discussion section.

Round 2
Reviewer 1 Report
The authors have addressed all comments adequately given the experimental limitations derived from working with PFFs.
Nevertheless, I would strongly recommend to attempt the experiment proposed in Major Comments Point 1, were after transfection gDNA is collected at day 2-3 without selection and assess editing. It would be the most suitable approach to measure editing efficiencies across editors. Nevertheless, I understand antibiotic selection is important to perform all experiments performed in the article afterwards.
Author Response
Response to Reviewer 1 Comments
The authors have addressed all comments adequately given the experimental limitations derived from working with PFFs.
Nevertheless, I would strongly recommend to attempt the experiment proposed in Major Comments Point 1, were after transfection gDNA is collected at day 2-3 without selection and assess editing. It would be the most suitable approach to measure editing efficiencies across editors. Nevertheless, I understand antibiotic selection is important to perform all experiments performed in the article afterwards.
Response: Following this comment, we performed a genome editing experiment without puromycin treatment using ABE8e and NG-ABE8e editors (showed superior efficiencies in the experiments with puromycin selection). The data showed that the editing efficiencies in non-selection groups were significantly lower (decreased by 50.04% and 32.82%) than those in puromycin selection groups. Nevertheless, the A to G editing efficiences were still impressive in the non-selection PFFs, suggesting that those ABE editors used in this study could be powerful tools for A to G editing in various mammalian cell lines. In the revision, we added a figure (Figure S6) and some descriptions in “Materials and Methods” and “Discussion” sections. For details, please see page 2, lines 93-96 and page 11, lines 338-341.

Reviewer 2 Report
Authors have addressed this reviewers comments sufficiently.
Author Response
We appreciate the positive comments from the reviewer.